# Explicit Regularization via Regularizer Mirror Descent

## Abstract

Despite perfectly interpolating the training data, deep neural networks (DNNs) can often generalize fairly well, in part due to the "implicit regularization" induced by the learning algorithm. Nonetheless, various forms of regularization, such as "explicit regularization" (via weight decay), are often used to avoid overfitting, especially when the data is corrupted. There are several challenges with explicit regularization, most notably unclear convergence properties. Inspired by the convergence properties of stochastic mirror descent (SMD) algorithms, we propose a new method for training DNNs with regularization, called regularizer mirror descent (RMD). In highly overparameterized DNNs, SMD simultaneously interpolates the training data and minimizes a certain potential function of the weights. RMD starts with a standard cost which is the sum of the training loss and a convex regularizer of the weights. Reinterpreting this cost as the potential of an "augmented" overparameterized network and applying SMD yields RMD. As a result, RMD inherits the properties of SMD and provably converges to a point "close" to the minimizer of this cost. RMD is computationally comparable to stochastic gradient descent (SGD) and weight decay and is parallelizable in the same manner. Our experimental results on training sets with various levels of corruption suggest that the generalization performance of RMD is remarkably robust and significantly better than both SGD and weight decay, which implicitly and explicitly regularize the $\ell_2$ norm of the weights. RMD can also be used to regularize the weights to a desired weight vector, which is particularly relevant for continual learning.

## 1 Introduction

### 1.1 Motivation

Today's deep neural networks are typically highly overparameterized and often have a large enough capacity to easily overfit the training data to zero training error (Zhang et al., 2016). Furthermore, it is now widely recognized that such networks can still generalize well despite (over)fitting (Bartlett et al., 2020; Belkin et al., 2018; 2019; Nakkiran et al., 2021; Bartlett et al., 2021), which is, in part, due to the "implicit regularization" (Gunasekar et al., 2018a; Azizan & Hassibi, 2019b; Neyshabur et al., 2015; Boffi & Slotine, 2021) property of the optimization algorithms such as stochastic gradient descent (SGD) or its variants. However, in many cases, especially when the training data is known to include corrupted samples, it is still highly desirable to avoid overfitting the training data through some form of regularization (Goodfellow et al., 2016; Kukačka et al., 2017). This can be done through, e.g., early stopping, or explicit regularization of the network parameters via weight decay. However, the main challenge with these approaches is that their convergence properties are in many cases unknown and they typically do not come with performance guarantees.

### 1.2 Contributions

The contributions of the paper are as follows.

**1**) We propose a new method for training DNNs with regularization, called regularizer mirror descent (RMD), which allows for choosing any desired convex regularizer of the weights. RMD leverages the implicit regularization properties of the stochastic mirror descent (SMD) algorithm. It does so by reinterpreting the

explicit cost (the sum of the training loss and convex regularizer) as the potential function of an "augmented" network. SMD applied to this augmented network and cost results in RMD.

**2**) Contrary to most existing explicit regularization methods, RMD comes with *convergence guarantees*, as a result of the connection to SMD. More specifically, for highly overparameterized models, it provably converges to a point "close" to the minimizer of the cost.

**3**) RMD is *computationally and memory-wise efficient.* It imposes virtually no additional overhead compared to standard SGD, and can run in mini-batches and/or be distributed in the same manner.

**4**) We evaluate the performance of RMD using a ResNet-18 neural network architecture on the CIFAR-10 dataset with various levels of corruption. The results show that the generalization performance of RMD is *remarkably robust to data corruptions* and significantly better than both the standard SGD, which implicitly regularizes the $\ell_2$ norm of the weights, as well as weight decay, which explicitly does so. Further, unlike other explicit regularization methods, e.g., weight decay, the generalization performance of RMD is very consistent and not sensitive to the regularization parameter.

**5**) An extension of the convex regularizer can be used to guarantee the closeness of the weights to a desired weight vector with a desired notion of distance. This makes RMD particularly relevant for *continual learning.*

Therefore, we believe that RMD provides a very viable alternative to the existing explicit regularization approaches.

### 1.3   Related Work

There exist a multitude of regularization techniques that are used in conjunction with the training procedures of DNNs. See, e.g., Goodfellow et al. (2016); Kukačka et al. (2017) for a survey. While it is impossible to discuss every work in the literature, the techniques can be broadly divided into the following categories based on how they are performed: (i) via *data augmentation*, such as mixup (Zhang et al., 2018b), (ii) via the *network architecture*, such as dropout (Hinton et al., 2012), and (iii) via the *optimization algorithm*, such as early stopping (Li et al., 2020; Yao et al., 2007; Molinari et al., 2021), among others.

Our focus in this work is on explicit regularization, which is done through adding a regularization term to the cost. Therefore, the most closely comparable approach is weight decay (Zhang et al., 2018a), which adds an $\ell_2$-norm regularizer to the objective. However, our method is much more general, as it can handle any desired strictly-convex regularizer.

Another related work is that of Hu et al. (2019), who proposed two different forms of regularization with convergence guarantees. Their first regularizer (RDI) is based on distance to initialization. This is a special case of our formulation (13) when $\psi(w) = \frac{1}{2}\|w\|^2$. Their second regularizer (AUX) is based on inserting an auxiliary variable inside the loss the function, i.e., $\frac{1}{2}\sum_{i=1}^{n}(f(w, x_i) + \alpha b_i - y_i)^2$. This is a different kind of regularizer from the standard augmented form and is thus not directly comparable with our approach.

As mentioned earlier, our algorithm for solving the explicitly-regularized problem leverages the "implicit regularization" behavior of a family of optimization algorithms called stochastic mirror descent (Azizan et al., 2021). We discuss this further in Section 2.3.

The rest of the paper is organized as follows. We review some preliminaries about explicit and implicit regularization in Section 2. We present the main RMD algorithm and its various special cases in Section 3. In Section 4, we perform an experimental evaluation of RMD and demonstrate its generalization performance. In Section 5, we show that RMD can be readily used for regularizing the weights to be close to any desired weight vector, which is particularly important for continual learning. We present the convergence guarantees of RMD in Section 6, and finally conclude in Section 7.

## 2   Background

We review some background about stochastic gradient methods and different forms of regularization.

### 2.1 Stochastic Gradient Descent

Let $L_i(w)$ denote the loss on the data point $i$ for a weight vector $w \in \mathbb{R}^p$. For a training set consisting of $n$ data points, the total loss is $\sum_{i=1}^n L_i(w)$, which is typically attempted to be minimized by stochastic gradient descent (Robbins & Monro, 1951) or one of its variants (such as mini-batch, distributed, adaptive, with momentum, etc.). Denoting the model parameters at the $t$-th time step by $w_t \in \mathbb{R}^p$ and the index of the chosen data point by $i$, the update rule of SGD can be simply written as

$$w_t = w_{t-1} - \eta \nabla L_i(w_{t-1}), \quad t \geq 1, \tag{1}$$

where $\eta$ is the so-called learning rate, $w_0$ is the initialization, and $\nabla L_i(\cdot)$ is the gradient of the loss. When trained with SGD, typical deep neural networks (which have many more parameters than the number of data points) often achieve (near) zero training error (Zhang et al., 2016), or, in other words, "interpolate" the training data (Ma et al., 2018).

### 2.2 Explicit Regularization

As mentioned earlier, it is often desirable to avoid (over)fitting the training data to zero error, e.g., when the data has some corrupted labels. In such scenarios, it is beneficial to augment the loss function with a (convex and differentiable) regularizer $\psi : \mathbb{R}^p \to \mathbb{R}$, and consider

$$\boxed{\min_w \ \lambda \sum_{i=1}^n L_i(w) + \psi(w),} \tag{2}$$

where $\lambda \geq 0$ is a hyper-parameter that controls the strength of regularization relative to the loss function. A simple and common choice of regularizer is $\psi(w) = \frac{1}{2}\|w\|^2$. In this case, when SGD is applied to (2) it is commonly referred to as weight decay. Note that the bigger $\lambda$ is, the more effort in the optimization is spent on minimizing $\sum_{i=1}^n L_i(w)$. Since the losses $L_i(\cdot)$ are non-negative, the lowest these terms can get is zero, and thus, for $\lambda \to \infty$, the problem would be equivalent to the following:

$$\begin{aligned} \min_w \quad & \psi(w) \\ \text{s.t.} \quad & L_i(w) = 0, \quad i = 1, \ldots, n. \end{aligned} \tag{3}$$

### 2.3 Implicit Regularization

Recently, it has been noted in several papers that, even *without* imposing any explicit regularization in the objective, i.e., by optimizing only the loss function $\sum_{i=1}^n L_i(w)$, there is still an implicit regularization induced by the optimization algorithm used for training (Gunasekar et al., 2018a;b; Azizan & Hassibi, 2019b; Lyu & Li, 2019; Poggio et al., 2020). Namely, when initialized at the origin, SGD with sufficiently small step size tends to converge to interpolating solutions with minimum $\ell_2$ norm (Engl et al., 1996; Gunasekar et al., 2018a), i.e.,[1]

$$\begin{aligned} \min_w \quad & \|w\|_2 \\ \text{s.t.} \quad & L_i(w) = 0, \quad i = 1, \ldots, n. \end{aligned}$$

More generally, it has been shown (Gunasekar et al., 2018a; Azizan et al., 2021) that SMD, whose update rule is defined for a differentiable strictly-convex "potential function" $\psi(\cdot)$ as

$$\nabla \psi(w_t) = \nabla \psi(w_{t-1}) - \eta \nabla L_i(w_{t-1}), \tag{4}$$

with proper initialization and sufficiently small learning rate tends to converge to the solution of[2]

$$\boxed{\begin{aligned} \min_w \quad & \psi(w) \\ \text{s.t.} \quad & L_i(w) = 0, \quad i = 1, \ldots, n. \end{aligned}} \tag{5}$$

---

[1]See Section 6 for a more precise statement.
[2]See Section 6 and Theorem 6.3 for a more precise statement.

Note that this is equivalent to the case of explicit regularization with $\lambda \to \infty$, i.e., problem (3).

## 3  Proposed Method: Regularizer Mirror Descent (RMD)

When it is undesirable to reach zero training error, e.g., due to the presence of corrupted samples in the data, one cannot rely on the implicit bias of the optimization algorithm to avoid overfitting. That is because these algorithms would interpolate the corrupted data as well. This suggests using explicit regularization as in (2). Unfortunately, standard explicit regularization methods, such as weight decay, which is simply employing SGD to (2), do not come with convergence guarantees. Here, we propose a new algorithm, called Regularizer Mirror Descent (RMD), which, under appropriate conditions, provably regularizes the weights for any desired differentiable strictly-convex regularizer $\psi(\cdot)$. In other words, RMD converges to a weight vector close to the minimizer of (2).

We are interested in solving the explicitly-regularized optimization problem (2). Let us define an auxiliary variable $z \in \mathbb{R}^n$ with elements $z[1], \ldots, z[n]$. The optimization problem (2) can be transformed into the following form:

$$\min_{w,z} \quad \lambda \sum_{i=1}^{n} \frac{z^2[i]}{2} + \psi(w) \tag{6}$$
$$\text{s.t.} \quad z[i] = \sqrt{2L_i(w)}, \quad i = 1, \ldots, n.$$

The objective of this optimization problem is a strictly-convex function

$$\hat{\psi}(w,z) = \psi(w) + \frac{\lambda}{2} \|z\|^2,$$

and there are $n$ equality constraints. We can therefore think of an "augmented" network with two sets of weights, $w$ and $z$. To enforce the constraints $z[i] = \sqrt{2L_i(w)}$, we can define a "constraint-enforcing" loss $\hat{\ell}\left(z[i] - \sqrt{2L_i(w)}\right)$, where $\hat{\ell}(\cdot)$ is a differentiable convex function with a unique root at 0 (e.g., the square loss $\hat{\ell}(\cdot) = \frac{(\cdot)^2}{2}$). Thus, (6) can be rewritten as

$$\boxed{\begin{aligned} \min_{w,z} \quad & \hat{\psi}(w,z) \\ \text{s.t.} \quad & \hat{\ell}\left(z[i] - \sqrt{2L_i(w)}\right) = 0, \quad i = 1, \ldots, n. \end{aligned}} \tag{7}$$

Note that (7) is similar to the implicitly-regularized optimization problem (5), which can be solved via SMD. To do so, we need to follow (4) and compute the gradients of the potential $\hat{\psi}(\cdot, \cdot)$, as well as the loss $\hat{\ell}\left(z[i] - \sqrt{2L_i(w)}\right)$, with respect to $w$ and $z$. We omit the details of this straightforward calculation and simply state the result, which we call the RMD algorithm.

At time $t$, when the $i$-th training sample is chosen for updating the model, the update rule of RMD can be written as follows:

$$\nabla \psi(w_t) = \nabla \psi(w_{t-1}) + \frac{c_{t,i}}{\sqrt{2L_i(w_{t-1})}} \nabla L_i(w_{t-1}),$$
$$z_t[i] = z_{t-1}[i] - \frac{c_{t,i}}{\lambda},$$
$$z_t[j] = z_{t-1}[j], \quad \forall j \neq i, \tag{8}$$

where $c_{t,i} = \eta \hat{\ell}'\left(z_{t-1}[i] - \sqrt{2L_i(w_{t-1})}\right)$, $\hat{\ell}'(\cdot)$ is the derivative of the constraint-enforcing loss function, and the variables are initialized with $w_0 = 0$ and $z_0 = 0$. Note that because of the strict convexity of the regularizer $\psi(\cdot)$, its gradient $\nabla \psi(\cdot)$ is an invertible function, and the above update rule is well-defined. Algorithm 1 summarizes the procedure. As will be shown in Section 6, under suitable conditions, RMD provably solves the optimization problem (2).

One can choose the constraint-enforcing loss as $\hat{\ell}(\cdot) = \frac{(\cdot)^2}{2}$, which implies $\hat{\ell}'(\cdot) = (\cdot)$, to simply obtain the same update rule as in (8) with $c_{t,i} = \eta\left(z_{t-1}[i] - \sqrt{2L_i(w_{t-1})}\right)$.

---

**Algorithm 1** Regularizer Mirror Descent (RMD)

---

**Require:** $\lambda, \eta, w_0$
 1: **Initialization:** $w \leftarrow w_0$, $z \leftarrow 0$
 2: **repeat**
 3:    Take a data point $i$
 4:    $c \leftarrow \eta \hat{\ell}' \left( z[i] - \sqrt{2L_i(w)} \right)$
 5:    $w \leftarrow \nabla \psi^{-1} \left( \nabla \psi(w) + \frac{c}{\sqrt{2L_i(w)}} \nabla L_i(w) \right)$
 6:    $z[i] \leftarrow z[i] - \frac{c}{\lambda}$
 7: **until** convergence
 8: **return** $w$

---

### 3.1 Special Case: $q$-norm Potential

An important special case of RMD is when the potential function $\psi(\cdot)$ is chosen to be the $\ell_q$ norm, i.e., $\psi(w) = \frac{1}{q}\|w\|_q^q = \frac{1}{q}\sum_{k=1}^p |w[k]|^q$, for a real number $q > 1$. Let the current gradient be denoted by $g := \nabla L_i(w_{t-1})$. In this case, the update rule can be written as

$$
\begin{aligned}
w_t[k] &= \left| \xi_{t,i} \right|^{\frac{1}{q-1}} \operatorname{sign}\left( \xi_{t,i} \right), \quad \forall k \\
z_t[i] &= z_{t-1}[i] - \frac{c_{t,i}}{\lambda}, \\
z_t[j] &= z_{t-1}[j], \qquad \forall j \neq i,
\end{aligned}
\tag{9}
$$

for $\xi_{t,i} = |w_{t-1}[k]|^{q-1} \operatorname{sign}(w_{t-1}[k]) + \frac{c_{t,i}}{\sqrt{2L_i(w_{t-1})}} g[k]$, where $w_t[k]$ denotes the $k$-th element of $w_t$ (the weight vector at time $t$) and $g[k]$ is the $k$-th element of the current gradient $g$. Note that for this choice of potential function, the update rule is *separable*, in the sense that the update for the $k$-th element of the weight vector requires only the $k$-th element of the weight and gradient vectors. This allows for efficient parallel implementation of the algorithm, which is crucial for large-scale tasks.

Even among the family of $q$-norm RMD algorithms, there can be a wide range of regularization effects for different values of $q$. Some important examples are as follows:

$\ell_1$ **norm** regularization promotes sparsity in the weights. Sparsity is often desirable for reducing the storage and/or computational load, given the massive size of state-of-the-art DNNs. However, since the $\ell_1$-norm is neither differentiable nor strictly convex, one may use $\psi(w) = \frac{1}{1+\epsilon}\|w\|_{1+\epsilon}^{1+\epsilon}$ for some small $\epsilon > 0$ (Azizan & Hassibi, 2019a).

$\ell_\infty$ **norm** regularization promotes bounded and small range of weights. With this choice of potential, the weights tend to concentrate around a small interval. This is often desirable in various implementations of neural networks since it provides a small dynamic range for quantization of weights, which reduces the production cost and computational complexity. However, since $\ell_\infty$ is, again, not differentiable, one can choose a large value for $q$ and use $\psi(w) = \frac{1}{q}\|w\|_q^q$ to achieve the desirable regularization effect of $\ell_\infty$ norm ($q = 10$ is used in Azizan et al. (2021)).

$\ell_2$ **norm** still promotes small weights, similar to $\ell_1$ norm, but to a lesser extent. The update rule is

$$
\begin{aligned}
w_t[k] &= w_{t-1}[k] + \frac{c_{t,i}}{\sqrt{2L_i(w_{t-1})}} g[k], \quad \forall k \\
z_t[i] &= z_{t-1}[i] - \frac{c_{t,i}}{\lambda}, \\
z_t[j] &= z_{t-1}[j], \qquad \forall j \neq i.
\end{aligned}
\tag{10}
$$

### 3.2 Special Case: Negative Entropy Potential

One can choose the potential function $\psi(\cdot)$ to be the negative entropy, i.e., $\psi(w) = \sum_{k=1}^{p} w[k] \log(w[k])$. For this particular choice, the associated Bregman divergence (Bregman, 1967; Azizan & Hassibi, 2019c) reduces to the Kullback–Leibler divergence. Let the current gradient be denoted by $g := \nabla L_i(w_{t-1})$. The update rule would be

$$
\begin{aligned}
w_t[k] &= w_{t-1}[k] \exp\left( \frac{c_{t,i}}{\sqrt{2L_i(w_{t-1})}} g[k] \right) \quad \forall k \\
z_t[i] &= z_{t-1}[i] - \frac{c_{t,i}}{\lambda}, \\
z_t[j] &= z_{t-1}[j], \qquad \forall j \neq i,
\end{aligned}
\tag{11}
$$

This update rule requires the weights to be positive.

## 4 Experimental Results

As mentioned in the introduction, there are many ways to regularize DNNs and improve their generalization performance, including methods that perform data augmentation, a change to the network architecture, or early stopping. However, since in this paper we are concerned with the effect of the learning algorithm, we will focus on comparing RMD with the standard SGD (which induces implicit regularization) and the standard weight decay (which attempts to explicitly regularize the $\ell_2$ norm of the weights). No doubt the results can be improved by employing the aforementioned methods with these algorithms, but we leave that study for the future since it will not allow us to isolate the effect of the algorithm.

As we shall momentarily see, the results indicate that RMD outperforms both alternatives by a significant margin, thus making it a viable option for explicit regularization.

### 4.1 Setup

**Dataset.** To test the performance of different regularization methods in avoiding overfitting, we need a training set that does not consist entirely of clean data. We, therefore, took the popular CIFAR-10 dataset (Krizhevsky & Hinton, 2009), which has 10 classes and $n = 50,000$ training data points, and considered corrupting different fractions of the data. In the first scenario, we corrupted 25% of the data points, by assigning them a random label. Since, for each of those images, there is a 9/10 chance of being assigned a wrong label, roughly $9/10 \times 25\% = 22.5\%$ of the training data had incorrect labels. In the second scenario, we randomly flipped 10% of the labels, resulting in roughly 9% incorrect labels. For the sake of comparison, in the third scenario, we considered the uncorrupted data set itself.

**Network Architecture.** We used a standard ResNet-18 (He et al., 2016) deep neural network, which is commonly used for the CIFAR-10 dataset. The network has 18 layers, and around 11 million parameters. Thus, it qualifies as highly overparameterized. We did not make any changes to the network.

**Algorithms.** We use three different algorithms for optimization/regularization.

1. Standard SGD (implicit regularization): First, we train the network with the standard (mini-batch) SGD. While there is no explicit regularization, this is still known to induce an implicit regularization, as discussed in Section 2.3.

2. Weight decay (explicit regularization): We next train the network with an $\ell_2$-norm regularization, through weight decay. We ran weight decay with a wide range of regularization parameters, $\lambda$.

3. RMD (explicit regularization): Finally, we train the network with RMD, which is provably regularizing with an $\ell_2$ norm. For RMD we also ran the algorithm for a wide range of regularization parameters, $\lambda$.

In all three cases, we train in mini batches—mini-batch RMD is summarized in Algorithm 2 in the Appendix.

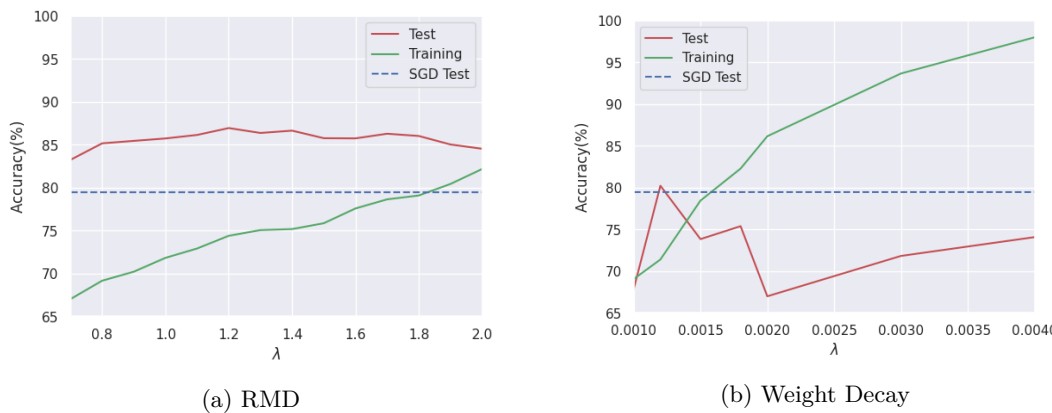

Figure 1: 25% corruption of the training set.

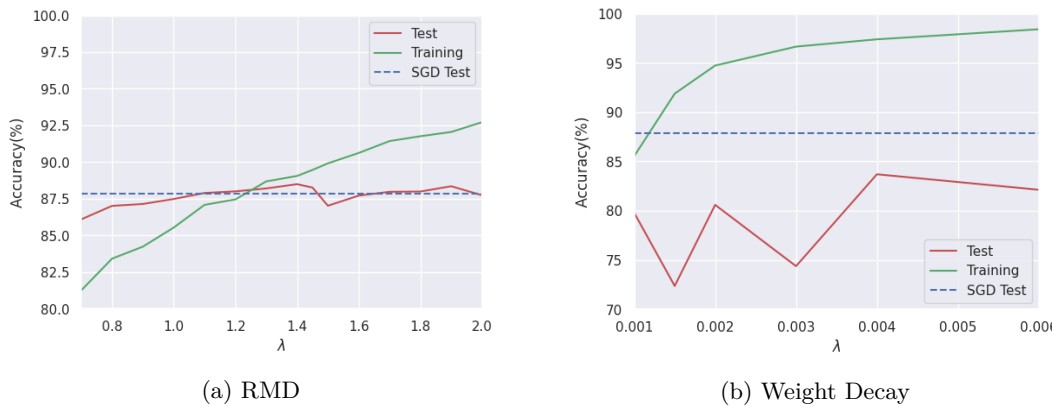

Figure 2: 10% corruption of the training set.

### 4.2 Results

The training and test accuracies for all three methods are given in Figs. 1-3. Fig. 1 shows the results when the training data is corrupted by 25%, Fig. 2 when it is corrupted by 10%, and Fig. 3 when it is uncorrupted.

As expected, because the network is highly overparameterized, in all cases, SGD interpolates the training data and achieves almost 100% training accuracy.

As seen in Fig. 1, at 25% data corruption SGD achieves 80% test accuracy. For RMD, as $\lambda$ varies from 0.7 to 2.0, the training accuracy increases from 67% to 82% (this increase is expected since RMD should interpolate the training data as $\lambda \to \infty$). However, the test accuracy remains generally constant around 85%, with a peak of 87%. This is significantly better than the generalization performance of SGD. For weight decay, as $\lambda$ increases from 0.001 to 0.004, the training accuracy increases from 70% to 98% (implying that there is no need to increase $\lambda$ beyond 0.004). The test accuracy, on the other hand, is rather erratic and varies from a low of 67% to a peak of 80%.

As seen in Fig. 2, at 10% data corruption SGD achieves 87.5% test accuracy. For RMD, as $\lambda$ varies from 0.7 to 2.0, the training accuracy increases from 82% to 92.5%. The test accuracy remains generally constant around and the peak of 88.5% is only marginally better than SGD. For weight decay, the training accuracy increases from 86% to 99%, while the test accuracy is erratic and peaks only at 80%.

Finally, for the sake of comparison, we show the results for the uncorrupted training data in Fig. 3. As expected, since the data is uncorrupted, interpolating the data makes sense and SGD has the best test

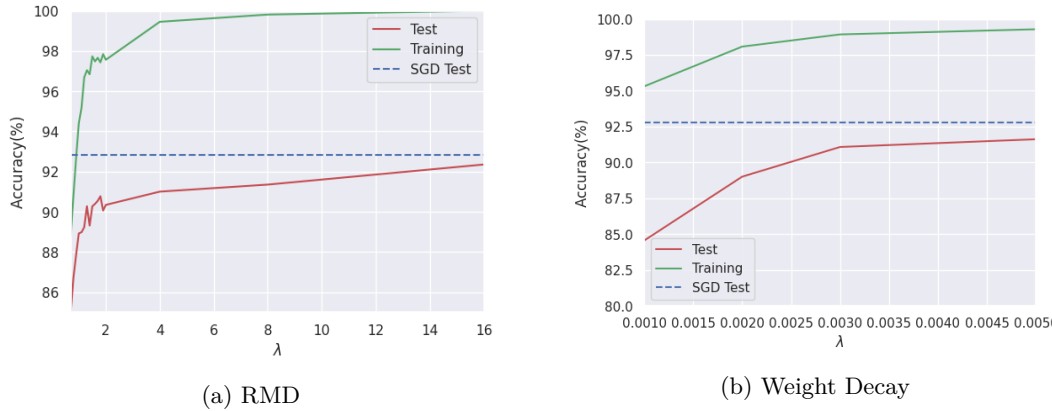

Figure 3: Uncorrupted training set.

accuracy. Both RMD and weight decay approaches have higher test accuracy as $\lambda$ increases, with RMD having superior performance.

We should also mention that we have run experiments with 40% corruption in the data. Here SGD achieves 70% test accuracy, while RMD achieves a whopping 81.5% test accuracy with only 64% training accuracy. See the Appendix for more details.

## 5 Regularization for Continual Learning

It is often desirable to regularize the weights to remain close to a particular weight vector. This is particularly useful for continual learning, where one seeks to learn a new task while trying not to "forget" the previous task as much as possible (Lopez-Paz & Ranzato, 2017; Kirkpatrick et al., 2017; Farajtabar et al., 2020). In this section, we show that our algorithm can be readily used for such settings by initializing $w_0$ to be the desired weight vector and suitably choosing a notion of closeness.

Augmenting the loss function with a regularization term that promotes closeness to some desired weight vector $w^{\mathrm{reg}}$, one can pose the optimization problem as

$$\min_{w} \quad \lambda \sum_{i=1}^{n} L_i(w) + \|w - w^{\mathrm{reg}}\|^2. \tag{12}$$

More generally, using a Bregman divergence $D_\psi(\cdot, \cdot)$ corresponding to a differentiable strictly-convex potential function $\psi : \mathbb{R}^p \to \mathbb{R}$, one can pose the problem as

$$\min_{w} \quad \lambda \sum_{i=1}^{n} L_i(w) + D_\psi(w, w^{\mathrm{reg}}). \tag{13}$$

Note that Bregman divergence is defined as $D_\psi(w, w^{\mathrm{reg}}) := \psi(w) - \psi(w^{\mathrm{reg}}) - \nabla\psi(w^{\mathrm{reg}})^T(w - w^{\mathrm{reg}})$, is non-negative, and convex in its first argument. Due to strict convexity of $\psi$, we also have $D_\psi(w, w^{\mathrm{reg}}) = 0$ iff $w = w^{\mathrm{reg}}$. For the choice of $\psi(w) = \frac{1}{2}\|w\|^2$, for example, the Bregman divergence reduces to the usual Euclidean distance $D_\psi(w, w_0) = \frac{1}{2}\|w - w^{\mathrm{reg}}\|^2$.

Same as in Section 3, we can define an auxiliary variable $z \in \mathbb{R}^n$, and rewrite the problem as

$$\min_{w,z} \quad \lambda \sum_{i=1}^{n} \frac{z^2[i]}{2} + D_\psi(w, w^{\mathrm{reg}})$$
$$\text{s.t.} \quad z[i] = \sqrt{2L_i(w)}, \quad i = 1, \dots, n. \tag{14}$$

It can be easily shown that the objective of this optimization problem is a Bregman divergence, i.e., $D_{\hat{\psi}}\left(\begin{bmatrix} w \\ z \end{bmatrix}, \begin{bmatrix} w^{\text{reg}} \\ 0 \end{bmatrix}\right)$, corresponding to a potential function $\hat{\psi}\left(\begin{bmatrix} w \\ z \end{bmatrix}\right) = \psi(w) + \frac{\lambda}{2}\|z\|^2$. As will be discussed in Section 6, this is exactly in a form that an SMD algorithm with the choice of potential function $\hat{\psi}$, initialization $w_0 = w^{\text{reg}}$ and $z_0 = 0$, and a sufficiently small learning rate will solve. In other words, Algorithm 1 with initialization $w_0 = w^{\text{reg}}$ provably solves the regularized problem (13).

## 6 Convergence Guarantees

In this section, we provide convergence guarantees for RMD under certain assumptions, motivated by the implicit regularization property of stochastic mirror descent, recently established in Azizan & Hassibi (2019b); Azizan et al. (2021).

Let us denote the training dataset by $\{(x_i, y_i) : i = 1, \ldots, n\}$, where $x_i \in \mathbb{R}^d$ are the inputs, and $y_i \in \mathbb{R}$ are the labels. The output of the model on data point $i$ is denoted by a function $f_i(w) := f(x_i, w)$ of the parameter $w \in \mathbb{R}^p$. The loss on data point $i$ can then be expressed as $L_i(w) = \ell(y_i - f_i(w))$ with $\ell(\cdot)$ being convex and having a global minimum at zero (examples include square loss, Huber loss, etc.). Since we are mainly concerned with highly overparameterized models (the interpolating regime), where $p \gg n$, there are (infinitely) many parameter vectors $w$ that can perfectly fit the training data points, and we can define

$$\mathcal{W} = \{w \in \mathbb{R}^p \mid f_i(w) = y_i, \ i = 1, \ldots, n\}$$
$$= \{w \in \mathbb{R}^p \mid L_i(w) = 0, \ i = 1, \ldots, n\}.$$

Let $w^* \in \mathcal{W}$ denote the interpolating solution that is closest to the initialization $w_0$ in Bregman divergence:

$$w^* = \arg\min_w \quad D_\psi(w, w_0)$$
$$\text{s.t.} \quad f_i(w) = y_i, \quad i = 1, \ldots, n. \tag{15}$$

It has been shown that, for a linear model $f(x_i, w) = x_i^T w$, and for a sufficiently small learning rate $\eta > 0$, the iterates of SMD (4) with potential function $\psi(\cdot)$, initialized at $w_0$, converge to $w^*$ (Azizan & Hassibi, 2019b).

When initialized at $w_0 = \arg\min_w \psi(w)$ (which is the origin for all norms, for example), the convergence point becomes the minimum-norm interpolating solution, i.e.,

$$w^* = \arg\min_w \quad \psi(w)$$
$$\text{s.t.} \quad f_i(w) = y_i, \quad i = 1, \ldots, n. \tag{16}$$

While for nonlinear models, the iterates of SMD do not necessarily converge to $w^*$, it has been shown that for highly-overparameterized models, under certain conditions, this still holds in an approximate sense (Azizan et al., 2021). In other words, the iterates converge to an interpolating solution $w_\infty \in \mathcal{W}$ which is "close" to $w^*$. More formally, the result from Azizan et al. (2021) along with its assumptions can be stated as follows.

Let us define $D_{L_i}(w, w') := L_i(w) - L_i(w') - \nabla L_i(w')^T(w - w')$, which is defined in a similar way to a Bregman divergence for the loss function. The difference though is that, unlike the potential function of the Bregman divergence, due to the nonlinearity of $f_i(\cdot)$, the loss function $L_i(\cdot) = \ell(y_i - f_i(\cdot))$ need not be convex (even when $\ell(\cdot)$ is). Further, denote the Hessian of $f_i$ by $H_{f_i}$.[3]

**Assumption 6.1.** Denote the initial point by $w_0$. There exists $w \in \mathcal{W}$ and a region $\mathcal{B} = \{w' \in \mathbb{R}^p \mid D_\psi(w, w') \le \epsilon\}$ containing $w_0$, such that $D_{L_i}(w, w') \ge 0, i = 1, \ldots, n$, for all $w' \in \mathcal{B}$.

**Assumption 6.2.** Consider the region $\mathcal{B}$ in Assumption 6.1. The $f_i(\cdot)$ have bounded gradient and Hessian on the convex hull of $\mathcal{B}$, i.e., $\|\nabla f_i(w')\| \le \gamma$, and $\alpha \le \lambda_{\min}(H_{f_i}(w')) \le \lambda_{\max}(H_{f_i}(w')) \le \beta, i = 1, \ldots, n$, for all $w' \in \text{conv } \mathcal{B}$.

---

[3]We refrain from using $\nabla^2 f_i$ for Hessian, which is typically used for Laplacian (divergence of the gradient).

**Theorem 6.3** (Azizan et al. (2021)). *Consider the set of interpolating solutions $\mathcal{W} = \{w \in \mathbb{R}^p \mid f(x_i, w) = y_i, \ i = 1, \ldots, n\}$, the closest such solution $w^* = \arg\min_{w \in \mathcal{W}} D_\psi(w, w_0)$, and the SMD iterates given in (4) initialized at $w_0$, where every data point is revisited after some steps. Under Assumptions 6.1 and 6.2, for sufficiently small step size, i.e., for any $\eta > 0$ for which $\psi(\cdot) - \eta L_i(\cdot)$ is strictly convex on $\mathcal{B}$ for all $i$, the following holds.*

    *1. The iterates converge to $w_\infty \in \mathcal{W}$.*

    *2. $D_\psi(w^*, w_\infty) = o(\epsilon)$.*

In a nutshell, Assumption 6.1 states that the initial point $w_0$ is close to the set of global minima $\mathcal{W}$, which arguably comes for free in highly overparameterized settings (Allen-Zhu et al., 2019), while Assumption 6.2 states that the first and second derivatives of the model are *locally* bounded. Motivated by the above result, we now return to RMD and its corresponding optimization problem.

Let us define a learning problem over parameters $\begin{bmatrix} w \\ z \end{bmatrix} \in \mathbb{R}^{p+n}$ with $\hat{f}_i\left(\begin{bmatrix} w \\ z \end{bmatrix}\right) = \sqrt{2L_i(w)} - z[i]$, $\hat{y}_i = 0$, and

$\hat{L}_i\left(\begin{bmatrix} w \\ z \end{bmatrix}\right) = \hat{\ell}\left(\hat{y}_i - \hat{f}_i\left(\begin{bmatrix} w \\ z \end{bmatrix}\right)\right) = \hat{\ell}\left(z[i] - \sqrt{2L_i(w)}\right)$ for $i = 0, \ldots, n$. Note that in this new problem, we now have $p+n$ parameters and $n$ constraints/data points, and since $p \gg n$, we have $p + n \gg n$, and we are still in the highly-overparameterized regime (even more so). Thus, we can also define the set of interpolating solutions for the new problem as

$$\hat{\mathcal{W}} = \left\{ \begin{bmatrix} w \\ z \end{bmatrix} \in \mathbb{R}^{p+n} \ \middle| \ \hat{f}_i\left(\begin{bmatrix} w \\ z \end{bmatrix}\right) = \hat{y}_i, \ i = 1, \ldots, n \right\}. \tag{17}$$

Let us define a potential function $\hat{\psi}\left(\begin{bmatrix} w \\ z \end{bmatrix}\right) = \psi(w) + \frac{\lambda}{2}\|z\|^2$ and a corresponding SMD

$$\nabla\hat{\psi}\left(\begin{bmatrix} w_t \\ z_t \end{bmatrix}\right) = \nabla\hat{\psi}\left(\begin{bmatrix} w_{t-1} \\ z_{t-1} \end{bmatrix}\right) - \eta\nabla\hat{L}_i\left(\begin{bmatrix} w_{t-1} \\ z_{t-1} \end{bmatrix}\right),$$

initialized at $\begin{bmatrix} w_0 \\ 0 \end{bmatrix}$. It is straightforward to verify that this update rule is equivalent to that of RMD, i.e., (8).

On the other hand, from (15), we have

$$\hat{w}^* = \underset{w,z}{\arg\min} \quad D_{\hat{\psi}}\left(\begin{bmatrix} w \\ z \end{bmatrix}, \begin{bmatrix} w_0 \\ 0 \end{bmatrix}\right)$$
$$\text{s.t.} \quad \hat{f}_i\left(\begin{bmatrix} w \\ z \end{bmatrix}\right) = \hat{y}_i, \quad i = 1, \ldots, n. \tag{18}$$

Plugging $D_{\hat{\psi}}\left(\begin{bmatrix} w \\ z \end{bmatrix}, \begin{bmatrix} w_0 \\ 0 \end{bmatrix}\right) = D_\psi(w, w_0) + \frac{\lambda}{2}\|z\|^2$ and $\hat{f}_i\left(\begin{bmatrix} w \\ z \end{bmatrix}\right) = \sqrt{2L_i(w)} - z[i]$ into (18), it is easy to see that it is equivalent to (14) for $w_0 = w^{\text{reg}}$, and equivalent to (6) for $w_0 = 0$. The formal statement of the theorem follows from a direct application of Theorem 6.3.

**Assumption 6.4.** Denote the initial point by $\begin{bmatrix} w_0 \\ 0 \end{bmatrix}$. There exists $\begin{bmatrix} w \\ z \end{bmatrix} \in \hat{\mathcal{W}}$ and a region $\hat{\mathcal{B}} = \left\{ \begin{bmatrix} w' \\ z' \end{bmatrix} \in \mathbb{R}^{p+n} \mid D_{\hat{\psi}}\left(\begin{bmatrix} w \\ z \end{bmatrix}, \begin{bmatrix} w' \\ z' \end{bmatrix}\right) \leq \epsilon \right\}$ containing $\begin{bmatrix} w_0 \\ 0 \end{bmatrix}$, such that $D_{\hat{L}_i}\left(\begin{bmatrix} w \\ z \end{bmatrix}, \begin{bmatrix} w' \\ z' \end{bmatrix}\right) \geq 0, i = 1, \ldots, n$, for all $\begin{bmatrix} w' \\ z' \end{bmatrix} \in \hat{\mathcal{B}}$.

**Assumption 6.5.** Consider the region $\hat{\mathcal{B}}$ in Assumption 6.4. The $\hat{f}_i(\cdot)$ have bounded gradient and Hessian on the convex hull of $\hat{\mathcal{B}}$, i.e., $\left\| \nabla \hat{f}_i \left( \begin{bmatrix} w' \\ z' \end{bmatrix} \right) \right\| \le \gamma$, and $\alpha \le \lambda_{\min} \left( H_{\hat{f}_i} \left( \begin{bmatrix} w' \\ z' \end{bmatrix} \right) \right) \le \lambda_{\max} \left( H_{\hat{f}_i} \left( \begin{bmatrix} w' \\ z' \end{bmatrix} \right) \right) \le \beta, i = 1, \ldots, n$, for all $\begin{bmatrix} w' \\ z' \end{bmatrix} \in$ conv $\hat{\mathcal{B}}$.

**Theorem 6.6.** *Consider the set of interpolating solutions $\hat{\mathcal{W}}$ defined in (17), the closest such solution $\hat{w}^*$ defined in (18), and the RMD iterates given in (8) initialized at $\begin{bmatrix} w_0 \\ 0 \end{bmatrix}$, where every data point is revisited after some steps. Under Assumptions 6.4 and 6.5, for sufficiently small step size, i.e., for any $\eta > 0$ for which $\hat{\psi}(\cdot) - \eta \hat{L}_i(\cdot)$ is strictly convex on $\hat{\mathcal{B}}$ for all i, the following holds.*

1. *The iterates converge to $\begin{bmatrix} w_\infty \\ z_\infty \end{bmatrix} \in \hat{\mathcal{W}}$.*

2. $D_{\hat{\psi}} \left( \hat{w}^*, \begin{bmatrix} w_\infty \\ z_\infty \end{bmatrix} \right) = o(\epsilon).$

Despite its somewhat complicated look, similar as in Assumption 6.1, Assumption 6.4 states the initial point $\begin{bmatrix} w_0 \\ 0 \end{bmatrix}$ is close to the (new) $(p + n)$-dimensional manifold $\hat{\mathcal{W}}$, which is reasonable because the new problem is even more overparameterized than the original $p$-dimensional one. Similar as in Assumption 6.2, Assumption 6.5 requires the first and second derivatives of the model to be locally bounded.

We should emphasize that while Theorem 6.6 states that we converge to the manifold $\hat{\mathcal{W}}$, it does *not* mean that it is fitting the training data points or achieving zero training error. That is because $\hat{\mathcal{W}} \in \mathbb{R}^{p+n}$ is a different (much higher-dimensional) manifold than $\mathcal{W} \in \mathbb{R}^p$, and interpolating it would translate to fitting the constraints defined by the regularized problem.

## 7 Conclusion and Outlook

We presented Regularizer Mirror Descent (RMD), a novel efficient algorithm for training DNN with any desired strictly-convex regularizer. The starting point for RMD is a standard cost which is the sum of the training loss and a differentiable strictly-convex regularizer of the network weights. For highly-overparameterized models, RMD provably converges to a point "close" to the minimizer of this cost. The algorithm can be readily applied to any DNN and enjoys the same parallelization properties as SGD. We demonstrated that RMD is remarkably robust to various levels of label corruption in data, and it outperforms both the implicit regularization induced by SGD and the explicit regularization performed via weight decay, by a wide margin. We further showed that RMD can be used for continual learning, where regularization with respect to a previously-learned weight vector is critical.

Given that RMD enables training any network efficiently with a desired regularizer, it opens up several new avenues for future research. In particular, an extensive experimental study of the effect of different regularizers on different datasets and different architectures would be instrumental to uncovering the role of regularization in modern learning problems.

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
