# OpenReview forum: "Explicit Regularization via Regularizer Mirror Descent"
_TMLR — Rejected by TMLR_

### Review · Reviewer_fycZ · 2022-06-05

**Summary Of Contributions:**

SGD is known to provide "implicit regularization" that benefits the generalization for deep nets. On the other hand, explicit regularizers, e.g., l2-regularization/weight-decay, provides explicit regularization. Motivated by these, this paper proposed a method called "regularized mirror descent" that combines the idea of SGD and explicit regularizers. Experiments under the setting of corrupted labels show that the proposed methods outperform the vanilla SGD or weight decay methods. Moreover, the paper studied the convergence of the methods based on existing results. Finally, the paper commented that the proposed methods may also be adopted in continual learning.

**Requested Changes:**

- revise the improper assessments that SGD with regu does not have theoretical analysis.

- improve the motivation of RMD

- revise the choice of $\lambda$

- demonstrate the effectiveness of RMD

**Strengths And Weaknesses:**

# Main questions
- In Sec 3 first paragraph, the paper claims that weight-decay or SGD does not come with convergence guarantees. This is simply not true. SGD with or without l2-regularization has been extensively studied in optimization/learning literature, the analysis for which could be found in even textbooks (for theory of optimization or machine learning). Even in the somewhat more modern settings, e.g., overparameterized NNs, SGD (+ l2 regu) is the default algorithm to be analyzed, see, e.g.,  [1] and references therein.

- In eq (2), the regularization parameter $\lambda$ is applied to the main loss instead of the regularization loss. While this is fine in theoretical considerations, this becomes problematic when running experiments, because even for the vanilla SGD (without regularization), $\lambda$ will affect the final output. Note that $\lambda$ will affect the effective learning rate for SGD. Therefore, I am confused by Figures 1, 2 and 3, where the SGD test error does not change when $\lambda$ varies.

- Similarly, $\lambda$ affects the effective learning rate for the proposed RMD and weight decay. I strongly suggest to apply $\lambda$ to the regularization term instead and report experimental results under this revision.

- I disagree with the comments for $\ell_1$-regu in Page 5. Note that although $\ell_1$ is not differentiable, $\ell_1$ is sub-differentiable. I do not see big issues for applying the vanilla $\ell_1$ regu in both implementation with tensorflow/pytorch or theoretical analysis.

- Page 6, Algorithms. How is weight decay method trained? By SGD? or Adam?
What is the $\hat{\ell}$ for RMD in the experiments?

- Page 7, Sec. 4.2 second paragraph. The authors discussed the reason for not considering a larger $\lambda$, which is a good remark. However the authors did not explain why not considering a smaller $\lambda$ than $0.001$. Note that $\lambda$ is applied to the main loss, thus a smaller $\lambda$ implies stronger regularization effect hence could potentially prevent overfitting in the label-corruption experiments.

- The presented theory seems to be direct consequence of existing results. Could you please explain the novelty here?

- In section 6 the authors wrote that "extensive experimental study...." I cannot agree that the presented set of experiments (Cifar-10 + resent + corrupted label) is "extensive".



## Summary
I feel the paper is not well motivated. The paper proposed "RMD" to combine the implicit and explicit regularization effects. However a more direct (and somehow already widely adopted) method is SGD + weight decay (or other regularizers). The paper argued such method "does not come with convergence guarantee" but this is not true.

The experiments are questionable due to the setup of $\lambda$. Moreover, Mixup (Zhang el al 2018b) and AUX (Hu et al 2019) are both known to be effective to mitigate the issue of label-corruption (there are more methods in literature). I do not think that the authors could ignore a comparison with these methods, even though the proposed method is based on a different motivation.

To sum up, I did not see sufficient reason to use "RMD" instead of SGD + regu, and as a consequence the contribution of this paper is limited in my perspective. I have to suggest a rejection to the paper in its current version.





[1] Cao, Yuan, and Quanquan Gu. "Generalization bounds of stochastic gradient descent for wide and deep neural networks." Advances in neural information processing systems 32 (2019).

---

### Review · Reviewer_UaCb · 2022-06-15

**Summary Of Contributions:**

This paper proposes regularizer mirror descent to explicitly regularize the weights of neural networks with strongly-convex regularizers. The authors claim that this is better than running gradient descent directly on regularized objectives.

**Broader Impact Concerns:**

The main contribution of the paper is a new algorithm for explicitly regularizing neural networks, so it is more on the methodology. I do not expect there to be any direct negative societal impact from this contribution.

**Requested Changes:**

Overall, I feel the biggest weakness of the paper is its motivation. I'm not convinced that the proposed algorithm is any better than explicitly regularizing the parameters. I hope the authors could clarify that.

In addition, I hope the authors could address my concerns that mentioned in the section of weaknesses.

**Strengths And Weaknesses:**

Strengths:
- The introduced algorithm looks interesting.

Weaknesses:
- The motivation of the paper is unclear. Why would the regularizer mirror descent algorithm be better than running gradient descent on regularized objectives? The authors argued in the paper that explicit regularization does not come with convergence guarantees. I don't think I would buy that. I don't see how adding L2 regularization would cause non-convergent issues.
- The authors discuss the implicit regularization of optimization algorithms multiple times in the paper. To my knowledge, stochastic optimizers enjoy two different types of implicit regularization, one from the optimization algorithm itself and another from the stochasticity. So I think the discussions should be made more accurate in the paper. Moreover, I believe gradient descent is minimizing the l2 norm to the initialization rather than to the origin.
- In the paper, the authors compared to explicit L2 regularization implemented with weight decay. However, it is known for modern architectures that L2 regularization is not equivalent to weight decay (see e.g. [1]). In this case, I don't think weight decay is the right baseline. In addition, for networks with batch normalization, the norm of weights has nothing to do with the predictions, which challenges the argument of this paper.
- In the paper, it was mentioned that explicit regularization of the network parameters does not come with any performance guarantee. I wonder if the author has any references. And also, do you have any performance guarantee for regularizer mirror descent other than the convergence guarantee in Section 6.
- The empirical results are quite weak and insufficient. Again, weight decay on ResNet18 has nothing to do with explicit regularization as ResNet18 uses batch normalization by default. Second, I don't understand why the training accuracy increases as you increase the weight decay regularization (in Figure 2).


References:
1. Three Mechanisms of Weight Decay Regularization.

---

### Review · Reviewer_1L94 · 2022-06-15

**Summary Of Contributions:**

This paper aims at designing an algorithm for solving explicitly regularized learning objectives, which are helpful to avoid overfitting, particularly in the case of corrupted data. The authors propose Regularizer Mirror Descent (RMD), which applies stochastic mirror descent to an "augmented" network and can be used when the regularizer is strictly convex. RMD enjoys convergence guarantees as well as efficiency in memory and computation. Besides use cases in avoiding overfitting, RMD can also be used in continual learning objectives. Empirically, RMD is compared to standard SGD and weight decays on the CIFAR-10 dataset with various levels of corruption, showing improvements in test accuracy as well as robustness to the regularization hyperparameter.

**Requested Changes:**

More empirical evaluation will make the paper stronger:
- Comparing the effect of RMD when combined with common techniques such as dropout.
- Comparing RMD with other regularization methods such as Hu et al 2019.
- Evaluating RMD for continual learning.

**Strengths And Weaknesses:**

**Strengths**

- The RMD algorithm is simple, efficient in memory and computation, and enjoys convergence guarantees.
- The mentioned strengths and promising empirical evaluation suggest that RMD may be helpful when combined with deep learning for handling corrupted data and in continual learning applications.
- The paper is well-written and easy to follow.

**Weaknesses**
Overall, the empirical evaluations are limited.
- It is true that comparison with standard SGD and standard weight decay allows for isolating the effect of the algorithm as the authors state. However, the current practice of deep learning commonly incorporates techniques such as data augmentation and dropout. Evaluating RMD when combined with these common techniques makes the result more relevant to the community.
- Empirical comparison only with standard SGD and l2 regularization is provided. The performance of RMD compared with other regularization methods is unclear.
- The paper explains how RMD can be used for continual learning but does not provide any theoretical or empirical results on that front.

---

### Review · Reviewer_5eic · 2022-06-15

**Summary Of Contributions:**

The paper provides a new (to my limited knowledge) interpretation of regularized training loss and then adapts classical stochastic mirror descent to yield a new method named RMD.

This view looks interesting to me and such a framework can be extended to the field of continual learning. However, it remains unclear if they can alleviate the catastrophic forgetting issue for uncorrupted training set.

**Requested Changes:**

please try to address the issues mentioned in the weaknesses section.

**Strengths And Weaknesses:**

# Strengths
* In general, the paper is well-structured, and easy to follow the idea of Regularizer Mirror Descent (RMD) in sections 2 & 3.
* Different regularization extensions of the proposed framework are provided.

# Weaknesses
1. The convergence results of RMD are not convincing, while the justifications/motivations look weak.
    * e.g. in the sentence "Inspired by the convergence properties of stochastic mirror descent (SMD) algorithms, we propose a new method for training DNNs with regularization", and the sentence "Contrary to most existing explicit regularization methods, RMD comes with convergence guarantees.". If I understand the content correctly, we have tight convergence rates for strongly convex functions with explicit regularization methods like the L2 norm, so authors are encouraged to better justify these sentences.
    * The reviewer is not familiar with the convergence proof of stochastic mirror descent, but the current results look trivial and have limited novelty.
2. Weak empirical results
    * The draft seems to completely omit the discussions and (empirically/theoretically) comparisons with existing methods proposed for learning with noisy labels, e.g., methods in [1, 2, 3]. Without considering some related methods therein, it is hard to identify the significance of the RMD.
    * Figure 2 and Figure 3 have different y-axis scales, making it very difficult to identify the difference/performance gain of the proposed method.
    * The value ranges of lambda for RMD and Weight decay are arbitrary/too small. Without examining a wider range of lambda, the empirical justifications are less-convincing.
    * The manuscript has spent at least 1-page introducing different regularization approaches, but only the l2 norm is evaluated in the experimental section. Some ablation studies should be provided here.
    * No empirical justification can be found for the effectiveness of extending RMD to continual learning.
3. Compatibility with the SOTA optimizers. The manuscript only considers mini-batch SGD as its baseline, while in practice, mini-batch SGD with Nesterov momentum or adaptive methods like Adam is widely used. Some comments/results on this are required, e.g., in terms of adapting RMD to mini-batch with Nesterov momentum.


# Reference
1. A Survey of Label-noise Representation Learning: Past, Present and Future
2. Image Classification with Deep Learning in the Presence of Noisy Labels: A Survey
3. Learning from Noisy Labels with Deep Neural Networks: A Survey

---

### Review · Reviewer_69nT · 2022-06-15

**Summary Of Contributions:**

The authors propose Regularizer Mirror Descent (RMD), an explicit regularization approach for training neural networks. RMD introduces auxillary variables $z[i]$ for each sample that enforces $z[i] = \sqrt{2L_i(\omega)}$, and solves the new implicitly-regularized problem with SMD. The authors show convergence result for the proposed algorithm, and compare  the algorithm with standard SGD and weight decay.



**Requested Changes:**

Please refer to the section on weakness.

**Strengths And Weaknesses:**

Strengths:

The proposed algorithm is novel and enjoys convergence property. And the empirical experiments on CIFAR-10 shows that the proposed method achieves better performance than standard SGD and weight decay.

Weakenss:

1. The main motivation of the proposed algorithm is to solve (2). And the authors claim that simply employing SGD to (2) does not come with convergence guarantees. I'm confused by the argument since as lambda goes to 0, the SGD shall enjoy some sort of convergence result (because in the limit of lambda = 0 the convergence is straightforward). Could the authors make this point more explicit? It would be great to show some seperation between RMD and standard SGD by creating a lower bound for the convergence rate of SGD in (2), or show an instance where SGD provably divergences.

2. In the experiments, the authors use standard SGD as a representative for implicit regularizations. I wonder how SMD compares with these methods, since SMD is closer to RMD (and as lambda goes to infinity they shall be the same). From the experiment results, it seems that a larger lambda in RMD does not hurt the test performance. If that's the case, why would we prefer RMD than SMD?

3. The motivation part is mainly presented in a single-sample setting, while the experimental results are mainly based on mini-batch algorithms. I wonder whether the intuition for implicit and explicit regularization generalizes to the case of mini-batch algorithms. Can the authors provide more comments on this?

4. Why do we take lambda in the range of [0.8, 2.0] for RMD, while only [0.001, 0.004] for weight decay? If as the authors claimed, they both aim to solve (2), then it appears weird to me why there is such a huge difference in how lambda affects the performance.

5. In the experimental results, RMD seems to enjoy an excellent generalization guarantee where the test accuracy can reach 85% even when the training accuracy is only 65%. This phenomenon does not happen to weight decay. I wonder whether similar phenomenon happens for SGD or SMD. And if not, why would we expect to see a much better test accuracy compared to training accuracy for RMD?

6. In terms of the theoretical analysis, the authors only show an asymptotic convergence result. Can the authors provide some comment on the convergence rate for RMD?

---

### Review · Reviewer_DZg3 · 2022-06-16

**Summary Of Contributions:**

This paper proposes RMD, a regularized version of mirror descent, which essentially introduces variables $z[i]$ for each sample $i$ in the dataset, and enforces $z[i] = \sqrt{2 L_i (w)}$. This constraint enables the objective to be strongly convex in the variables and allows writing the problem as equivalent to one which is solvable by mirror descent with implicit regularization. The authors extend the analysis of Azizan et al to give a convergence guarantee for RMD. With corrupted labels, the authors show that the proposed methods outperform vanilla SGD and weight decay method. The authors also remark that this algorithm may be used for continual learning.

**Requested Changes:**

Refer above.

**Strengths And Weaknesses:**

The paper is written in a simple and easy to read language. However, there are many weaknesses which I point out below.

Weaknesses:

- In point (2) of the contributions of the paper, the authors state that: "Contrary to most existing explicit regularization methods,  RMD comes with convergence guarantees, as a result of the connection to SMD". This statement comes off as ignorant of a large body of work studying L1/L2 regularized gradient descent in many different settings, both convex and non-convex.

- There is no discussion about how SMD with implicit regularization compares with RMD on the experiments in the paper, which seem to suggest that larger lambda improves test performance. But in this regime, the algorithm resembles SMD more and more closely?

- The method seems to basically be doing a change of variables for optimization. The novelty is very limited in this regard. Moreover, the convergence analysis largely follows from previous papers.

- How does one reconcile the fact that the range of lambda for RMD and for weight decay in the experiments have such different scales?

- It would help if the authors could comment on the finite-sample convergence rate for RMD in comparison with SGD with implicit regularization. It is unclear to me that the algorithm should converge any faster.

- The discussion of the algorithm in the context of continual learning is lacking. There are no theories or experiments to support the claims made in the paper regarding the expectation of superior performance of RMD in this setting.

In conclusion, I think that the novelty in the paper is very limited, the theory is weak and largely follows from previous work, and the experiments are not at all comprehensive. I do not recommend the paper for acceptance as it currently stands. Perhaps with more comprehensive experiments and more theoretical/practical justifications to believe that the algorithm performs better than SGD with implicit regularization, the paper may become stronger.

---

### Decision · Action_Editors · 2022-07-17

**Recommendation:** Reject

**Comment:**

The reviewers unamimously agree that the paper may not be accepted as it stands now. In particular, the theory is not strong and largely follows from previous work, and the experiments are not comprehensive. In particular, the paper lacks good comparison with existing work and convincing demonstration showing that it is superior.